# Autonomous design of new chemical reactions using a variational autoencoder

Robert Tempke[1] & Terence Musho [1✉]

Artificial intelligence based chemistry models are a promising method of exploring chemical reaction design spaces. However, training datasets based on experimental synthesis are typically reported only for the optimal synthesis reactions. This leads to an inherited bias in the model predictions. Therefore, robust datasets that span the entirety of the solution space are necessary to remove inherited bias and permit complete training of the space. In this study, an artificial intelligence model based on a Variational AutoEncoder (VAE) has been developed and investigated to synthetically generate continuous datasets. The approach involves sampling the latent space to generate new chemical reactions. This developed technique is demonstrated by generating over 7,000,000 new reactions from a training dataset containing only 7,000 reactions. The generated reactions include molecular species that are larger and more diverse than the training set.

[1] Department of Mechanical & Aerospace Engineering, West Virginia University, Morgantown, WV 26525, USA. ✉email: tdmusho@mail.wvu.edu

Artificial Intelligence has become one of the most important tools in the scientific community; it is used for everything from fundamental characterization to real-world applications. This is especially true in the field of chemistry and chemical engineering, where machine learning, a subset of artificial intelligence, has been used in conjunction with density functional theory (DFT), as well as on its own, to produce unique insight and provide rapid results[1,2]. Machine Learning has also been used in the chemical engineering field to help search the solution space of possible reactions to help save both time and resources[3–7]. With the increased use of machine learning in chemistry, researchers have begun to see that a biased problem exists in every available dataset of chemical reactions[8–10]. Most chemistry datasets come from a collection of patents and research articles that exist on the internet, however, this does not complete the continuous chemical reaction solution space[9,11]. This is often a sparse representation of the space. This study aims to utilize the predictive abilities of deep learning to synthetically generate a chemical reaction dataset that is less biased and more robust than the ones currently available or that can be data mined. This is accomplished using a generation deep learning technique known as a variational autoencoder (VAE)[12]. It is hypothesized that a VAE will form a sort of custom chemical compression intelligence that will provide efficient generation of new reactions by sampling the latent space of the VAE. This is a unique approach that relies on artificial intelligence to generate new reactions rather than retrosynthesizing[5–7] or breaking apart reactions and tracking the molecular species.

The issue with biased datasets in the teaching of machine learning algorithms comes down to a common principle of inherited bias evident in most artificial intelligence techniques, sometimes colloquially known as, "garbage in, garbage out"[13]. Having good data is the most important aspect of any artificial intelligence technique. However, good data is not just having optimized data or accurate data. Machine learning techniques do much better when taught on a range of inputs and outputs and the space is continuous[13,14]. In essence, a machine learning technique needs to learn everything, so it can know everything. In the paper by Jia et al., a detailed explanation is given on how researchers' biases go into the creation and design of almost all experiments. They go on to show how this can be combined with other types of biases to skew reported data[9]. Their work is further backed up by the Griffiths et al. study of biases in the natural sciences. A detailed explanation of this influence is based on the effects of data splitting noisy datasets as well as the influence that contextual variables can have on the outcome of experiments[8]. Kovács et al. do an excellent job of illustrating the direct effects that a biased and unbiased dataset can have on the quality of a machine learnings' outputs[10]. All of these different studies combine to show that the current methodology for creating datasets from publications or other open-source resources is flawed and will limit the ability of future machine learning algorithms. Glavatskikh et al. explains how the lack of diversity in data limits machine learning potential to predict[15]. Meanwhile, the cost of building an unbiased, continuous dataset is potentially an infinite feat when everything must be learned to know everything.

Despite the limited dataset availability and potential biases in datasets, research is currently demonstrating the employment of artificial intelligence techniques in the field of reaction chemistry. In the organic chemistry field, Kayala, and Baldi demonstrated how machine learning can be utilized to predict multistep reactions that take place over a range of reactants and thermodynamic conditions[4]. Moreover, they demonstrate how machine learning can be employed despite the limitations of the training dataset, while still being able to predict mechanistic reactions. In their study, they discuss the necessity of manually sorting the dataset

for the reactions they could utilize[16]. A follow-on study by Kovács et al. dissected their paper and concluded that the Clever Hans effect was apparent. This is essentially a failure of the double-blind condition. It was claimed that the correct prediction of the reactions was achieved but only because of a resulting reaction bias[10]. Similarly, Mater and Coote emphasize the common problem of bias in chemical reaction datasets[17]. These examples illuminate that while it is understood that bias plays an important role in the predictions of machine learning, no consensus has been reached on how to deal with these issues stemming from datasets.

However, in many different fields that face similar dataset issues, the use of generational techniques has shown excellent progress. The most famous examples of these are typically found in the medical field, where it is notoriously difficult to share patient data (due to Health Information Privacy) amongst scientists and researchers. Choe et al. devised a way to generate realistic synthetic data that contained high-dimensional discrete variables[18]. Their network, known as MedGAN, has shown excellent comparative performance to real-world datasets. Rigorous analysis has been done on the generated datasets, including distribution statistics, predictive modeling tasks, as well as expert review[18]. These findings are backed up by the research of Camino et al. and Gulrajani et al., who demonstrated that generative networks can be used to generate multi-categorical outputs completely synthetically[19,20]. This is an important aspect of generational techniques, especially VAEs. It demonstrates the autonomous synthetic generation of data for both discrete and continuous variables[21,22].

A popular generative technique is known as generative adversarial networks (GAN), as demonstrated by Choe et al. in MedGAN. These have notably outperformed VAE on various use cases, such as image generation and discrete variable representations of problems[23]. However, they are notoriously hard to work with, resulting in very unstable networks. More importantly, GAN needs a lot of data and a lot of tuning. This makes it undesirable for smaller datasets typically found in experimental chemical reaction datasets. In addition, there is not a latent space generation when using a GAN. This limits the usability in future use cases such as using latent space as an input into a generational network[24].

Variational autoencorders are a very active area of research where Yu et al. have demonstrated that generative networks can be used to model long sequences that rely on earlier segments to be semantically correct[25]. More important, generative networks have been shown to be able to generate missing data within a dataset[26]. This means that these deep learning techniques can interpret between existing data points to generate new synthetic-derived points that share the same characteristics as the original dataset.

One approach, and a competing approach to the proposed method, that has been recently employed for the generation of chemical reactions is the approach of retrosynthesizing known reactions[5–7]. The idea of retrosynthizing involves subsequently breaking the reaction products down by putting them back into the reaction process as reactants. This provides a complete continuous space of intermediate molecular species, but it is limited on the generation of new reaction molecular species that are not subspecies of the initial species database. Chemist have been taking retrosynthesis farther by combining artificial intelligence into the process. The work by Segler et al. show a promising route of combining Monte Carlo tree search with symbolic artificial intelligence. Which leds to a speed up of 30 times over traditional methods[27].

Other work has been done by several research groups that focus on improving the outcome of chemical reaction predictions.

Research by Shields et al. shows how Bayesian optimization can be used to fine-tune neural networks in synthetic chemistry[28]. In their research, they demonstrate how Bayesian optimization can essentially be thought of as an autonomous tool for limiting human biases. They then go on to demonstrate how removing expert chemists and engineers in real-life experiments and instead of using autonomous optimization of experiments they can get both a higher optimization efficiency and a better consistency. However, the Bayesian optimization only optimizes parameters that a human has set and cannot create new parameters as it learns.

A review of reaction prediction methods by Gale and Durand shows how almost all areas of machine learning in chemistry are in need of improvement and are active areas of research[29]. Some of the important topics they touch on are the need for datasets to have not only error-free reactions but also negative results. They discuss the difficulty in encoding chemical information for a machine-readable format. Another key point in their discussion is how you can use generative, discriminative deep neural networks to assess chemical reaction synthesis. Unfortunately, there is no discussion on how latent representations of chemical data can be used to generate feasible reaction synthesis.

Research by Iovanac et al. is a great example of how an encoder and decoder style neural network can be used to represent a continuous chemical latent space. Their research uses both experimental and density functional theory predicted models to predict properties of various $pK_a$ prediction of moderately sized molecular species[30]. However, this is just an illustrative example, what they clearly show is how scarce data can be overcome in the chemical space by using a latent representation of chemical molecules. They even demonstrate how that latent space can be used as training data for a neural network. This study takes this work a step farther and expands this methodology not only to single molecules but to whole equations containing a variety of molecules of different sizes. Unlike previous studies that focus on reactants predicting products or vice versa, this study instead allows for the latent representation of the data to predict both parts of an equation.

Other methods found in literature have demonstrated the ability of neural networks to autonomously predict chemical attributes of equations. Take the research from Zhang et al. where they use a combination of unsupervised $K$-means clustering and support vector machines to help in the prediction of activation energy of catalytic chemical reactions. They use the outputs of these unsupervised methods as inputs to a trained neural network. This is another way of generating a latent space representation of the dataset[31]. In comparison to the study by Iovanac et al. the $K$-means clustering, and the classification can be thought of as the encoder while the predictive neural networks would be the decoder. The focus of Zhang et al. research was on using the whole chemical reaction and their physical quantities to predict the activation energy.

This study introduces a method referred to as AGoRaS for the synthetic production of large quantities of balanced chemical reactions that can be utilized for the unbiased training of artificial intelligence techniques. It can also help in generating new targeted reactions, which can help guide experimental studies. Previous research by Amini et al. demonstrated how VAE can be used to generate unbiased synthetic machine learning training data for the training of more robust and accurate algorithms[32]. The following study proves that VAEs can be applied to chemistry to generate a large number of both new reactions and new species. This will allow not only for the improvement of machine learning algorithms but also offer significant time and cost savings to experimental studies. The ability of these networks to generate a near-continuous and infinite space of new reactions will allow for researchers to sort the data for specific products and reactants for a given thermodynamics condition.

## Results

**AGoRaS machine learning model**. This study utilized the probabilistic sampling of the latent space by VAE to explore the solution space for a dataset of gas-phase reactions. The model VAE model that was created was named AGoRaS. The VAE procedure employed is illustrated in Fig. 1A. Once the VAE is trained the new reactions can be generated by sampling the latent space, as illustrated in Fig. 1B. The only information given to AGoRaS-VAE is the encoded SMILES string and the only output is an encoded SMILES string, in which hydrogen atoms are implicit. The VAE approach can be thought of as a custom compression technique for these chemical reactions. Moreover, the latent space can be thought of as the memory of artificial intelligence. While it is not apparent by looking at the weights of the system, the artificial intelligence is empirically learning about the underlying physics and chemistry of the process. At the very essence, this is why it is necessary to remove bias. With bias present, there is a skew in the memory of the network. A unique approach applied in this research is that instead of each latent variable being encoded with a discrete value, as with a traditional autoencoder, instead, the latent variable has an associated probability distribution. Once trained, the probability distributions of each variable can be sampled to generate new output features or chemical equations. By using continuous values instead of a discrete value it forces a smooth latent space representation of the data, Gaussian in nature. In the case of AGoRaS, each SMILES character is represented uniquely as a digit and the entire equation string is encoded. By sampling different points along the latent variable Gaussian curves, a new reaction can be generated. While this new reaction is not guaranteed to be balanced, it is necessary to check for compliance. Once trained, AGoRaS can generate new reactions quickly with the gained knowledge of the input dataset.

**Validation workflow for AGoRaS**. Due to the inherently complex nature of chemical reactions, as well machine learning algorithms' susceptibility error generation based on training data and trained knowledge, it was important to set up a stringent pipeline or workflow for the training, validation, and analysis of the chemical reactions. To develop the training dataset, it was critical that the dataset being used for training contained molecular species could be converted to the SMILES format. This ensured that each chemical reaction in the dataset was unique and could be balanced. Each species contained in the training dataset was also checked in RDKit[33] and PubChem[34]. Once trained, the network was continually sampled to generate hundreds of thousands of equations. These equations were then fed through a series of validation steps using both RDKit, PubChem, and semi-empirical modeling software[33–35] to ensure validity and stability. These steps were used to validate that the equations were balanced, and the SMILES species represented physically possible chemical species. Figure 2 illustrates a simplified flow chart of the steps needed to accomplish the training and validation of AGoRaS. The semi-empirical modeling technique used the generated SMILES notation that is converted to an atomistic description to predict the thermodynamics properties.

To allow for the independent validation of the network, it was decided to use an easily accessible dataset for training. To that end, the NIST chemical kinetics database was selected based on being open access and well-documented sources[36]. Since the dataset is comprised of solely published reactions, it was determined that it would make an excellent case study of the

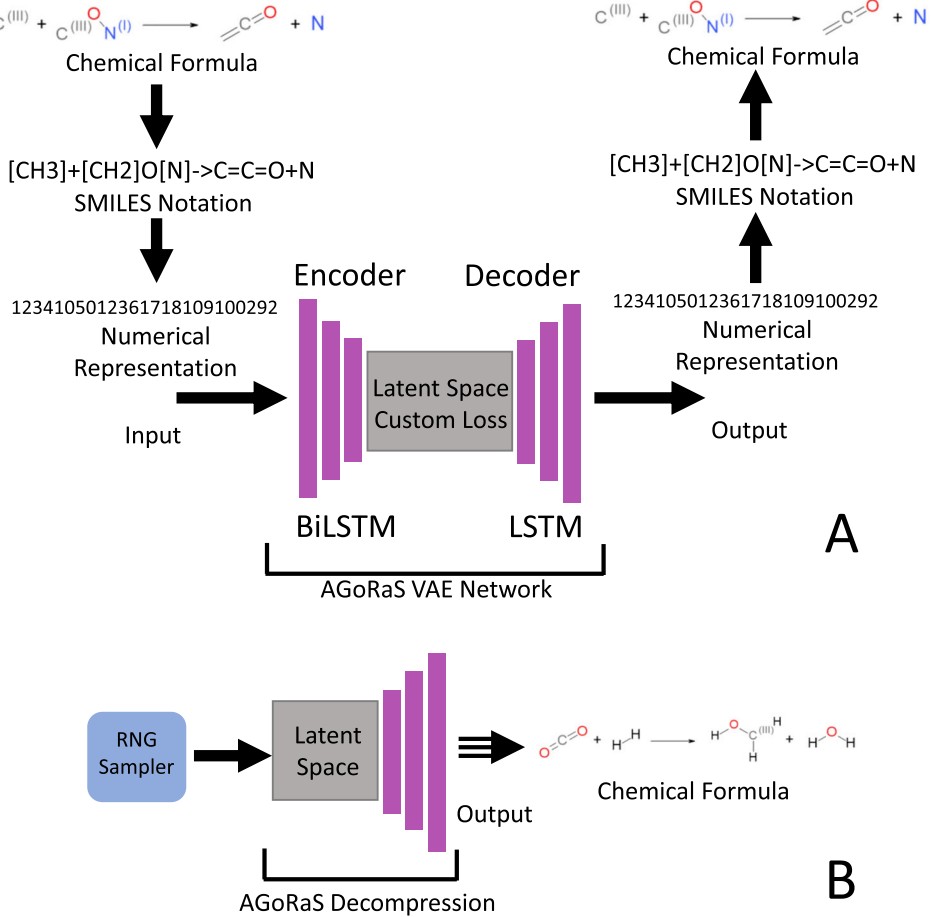

**Fig. 1 Workflow of the AGoRaS-based VAE network.** As illustrated in (**A**), the chemical database information is compressed and decompressed to form a high-dimensional latent space. **B** illustrates the workflow of how the trained latent space is sampled to generate new chemical equations.

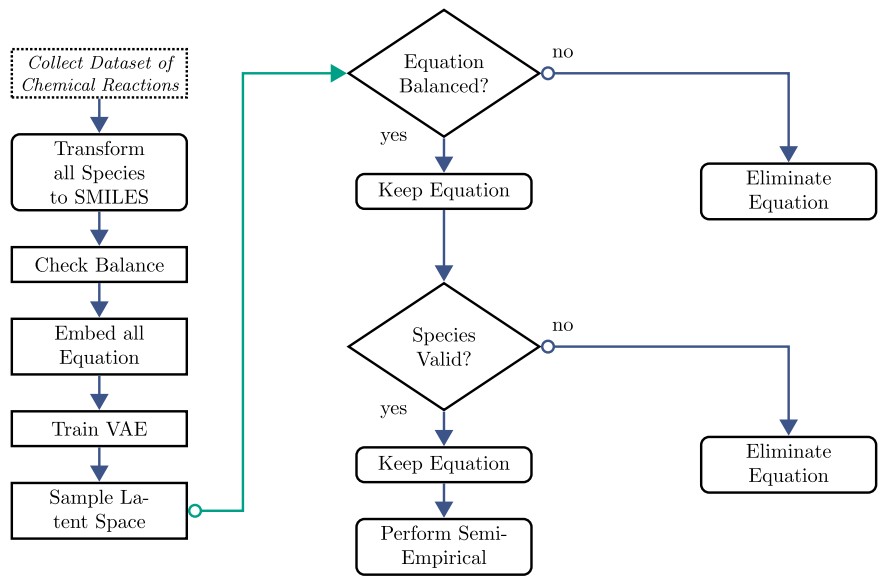

**Fig. 2 Flow diagram of the data collection, training, and validation steps taken by AGoRaS to generate synthetic data.** The generated equation is checked for both balance and existence. The semi-empirical calculation receives a SMILES descriptor and conducts an independent series of processes to generate a clean atomistic description prior to calculation of thermodynamics data.

ability of AGoRaS to generate a wide variety of equations from a sparse dataset. After data collection and cleaning a core number of 7000 chemical reactions, this resulted in ~2000 unique gas-phase molecular species.

**Validating generated equations**. Once the original dataset had been created and cleaned the VAE was trained, and the latent space was sampled. Sampling was stopped once 7,000,000 valid equations had been disseminated. This was selected as an

arbitrary stopping criteria of 1000 times the size of the original dataset. Among these new equations, AGoRaS was also able to generate ~20,000 new species, with both subspecies and completely new molecular species. The ability to predict new and larger molecular species owes to the success of this approach over other approaches, such as retrosynthesizing. All species were checked using the validation pipeline to ensure their uniqueness and stability.

To apply utility to the generated dataset and check stability of individual molecular species, the Gibbs free energy of each molecular species, along with the overall reaction energy, were determined at the standard state using a semi-empirical computational technique[35,37]. Furthermore, the difference in the dipole moments of the overall reactions was also predicted. The Gibbs free energy was selected because it is a good quantitative measure of the thermodynamics of the reaction. The dipole moment was also selected because of its thermodynamic relationship and application to electrocatalysis.

To calculate the Gibbs free energy of each reaction, the individual molecular species' entropy and enthalpies were calculated. The Gibbs energy of the reaction is defined as the following,

$$\Delta G^o_{reaction} = \Delta G^o_{products} - \Delta G^o_{reactants}, \tag{1}$$

where the Gibbs energy of the reactant and products are based on the following relationship:

$$\Delta G^o = \Delta H^o + T\Delta S^o. \tag{2}$$

Here the change in enthalpy ($\Delta H$) is based on the total energy of the semi-empirical calculation that is calibrated to the standard state in the semi-empirical calculation. The zero-point energy in this case is encapsulated within the enthalpy. The change in entropy ($\Delta S$) is based on the vibrational frequencies, moments of inertia of the molecule, and its symmetry number.

Similarly, the difference in entropy is calculated between the product and reactant species as follows:

$$\Delta S^o_{reaction} = \Delta S^o_{products} - \Delta S^o_{reactants}, \tag{3}$$

where the entropy terms ($\Delta S^o_{p,r}$) are corrected for the standard state. While this metric is encompassed in the overall Gibbs energy, it provides quantitative values for the amount of energy that is contributing to vibrational energy. Moreover, it provides a quantitative measure of the size of the molecules.

While more advanced, first principle DFT calculations are potentially desired in this application, the computational expense of 7,000,000 simulations outweighs the accuracy required for this study. The assumed approach will be to conduct a higher fidelity modeling approach for promising reactions, when applied to practice.

Due to the reporting bias in the original dataset, selection of best reaction, or thermodynamically favorable reactions (±1 eV) was known to be skewed towards reactions with low reaction energies. This was assumed to generate a bias in the network. However, it was found that generated equations ~97% fell within a $\Delta G_o$ of ±5 eV. This is important for the practical use cases of AGoRAS, as it shows that even without providing any context on thermodynamic properties during training the network is able to produce stable results over a much larger range than the training data.

The dipole moment, was also derived from the semi-empirical calculation. The dipole moment is calculated based on the partial charge of the atoms and the position of the atom. The following expression is used in the calculation:

$$\mu = \sum_{a=1}^{N} q_a r_a, \tag{4}$$

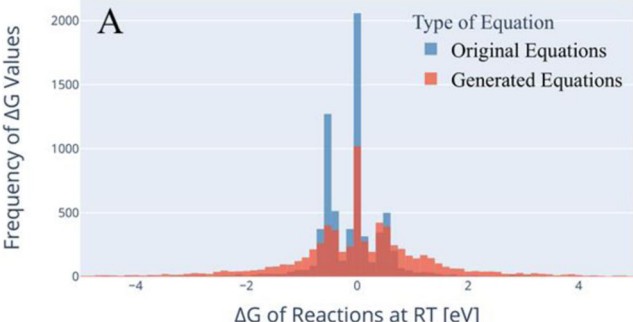

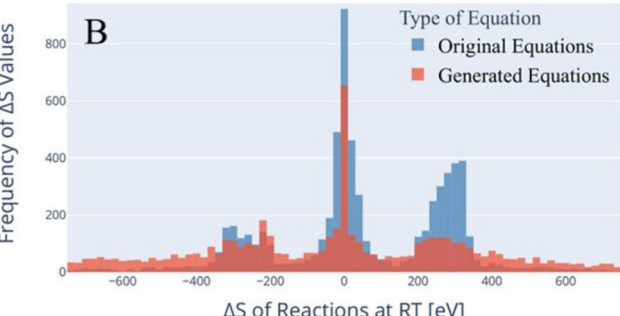

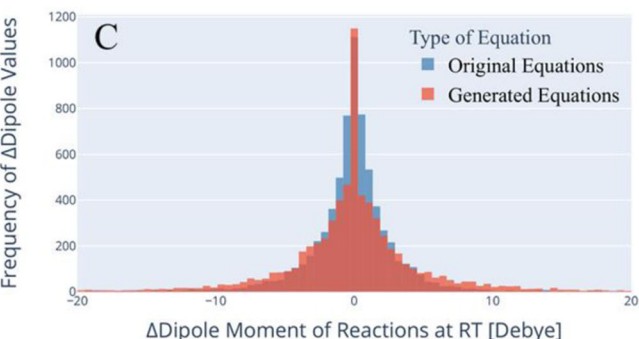

**Fig. 3 Histograms of the thermodynamics values comparing the training data (original equations) and the generated data (generated equations).** **A** is the Gibbs energy associated with Eq. (1). **B** is the difference in entropy between the product and reactant molecular species. **C** is the difference between the product and reactant dipole moment.

where $N$ is the total number of atoms, $q_{a,i}$ is the partial charge of atom $a$, and $r_{a,i}$ is the position vector of the atom $a$. In this study, the difference between the product and reactant dipole moments were calculated using the following:

$$\Delta\mu = \mu_{products} - \mu_{reactants}. \tag{5}$$

To visually and statistically compare trained and generated datasets of vastly different sizes, a random sample of 7000 equations was subselcted out of the generated dataset. An overlapping comparison between datasets was performed to confirm a correlation between the two datasets. This comparison is shown in Fig. 3, where A is a plot of the Gibbs free energy, B is a plot of the entropy, and C is a plot of the dipole moment. In all of these plots, the training dataset is pictured in blue in the background with the generated dataset in the foreground in red. The y-axis in all the plots is the frequency of occurrence and the x-axis is the associate thermodynamic property. It should be noted that 7,000,000 reaction equations were generated and this is a flat random selection of those generated. In subfigure plots A and B, there is a noted tri-modal distribution, which is a result of the tri-modal distribution of the training data. This is evidence of

the inherited bias in the training data that is being used. The AGoRaS is picking up on this knowledge as it generates a similar tri-modal distribution. The important region is the regions between and outer peaks of the input data. The AGoRaS is able to fill in these regions, which are either new or intermediate reactions. This generation ability is not based on retrosynthesizing previous reactions, but rather on the knowledge it has gained during training of the latent space variables. It can be shown, if all seven million generated reactions are plotted, that the distribution becomes more continuous.

Another interesting finding is apparent in the entropy plot of Fig. 3B. It is noted in this figure that AGoRaS has an flat distribution below −400 eV and above 400 eV. As seen in Eq. (3), a negative difference means the entropy is greater on the reactant side, or smaller on the product side. A thorough inspection of the generated dataset confirms that AGoRaS is generating larger molecules species on the reactant side. The size of these new species is beyond the size (number of atoms) of the input molecules species in the training dataset. This is a significant advantage of this approach, the ability to generate larger molecular species. Likewise, in Fig. 3A, there is bias arising from the training data, but the AGoRaS can fill in between the tri-modal peaks and extend beyond. It is also able to generate larger species on the product size as demonstrated with large positive values in Fig. 3B.

Figure 3C is a plot of the difference in dipole defined by Eq. (5). From a molecular point of view, this has the opposite trend of the entropy, as the molecule get larger (increased number of atoms) it is more likely the charge will be neutral for the molecule and have a corresponding negligible dipole moment. Therefore, as seen in Fig. 3, both the original and generated datasets have a single peak near zero. This means most molecular species have near-zero dipole moments, as expected. Moreover, the generated dataset contains some cases which have large negative differences in dipole moment. This means the reactants have a larger dipole moment than the products. This is significant because, based on Eq. (4), the position and charge of the atoms are critical to the dipole. This alludes to the fact that the AGoRaS has the ability to place atoms with ionic bonding tendencies further out from the center of charge, resulting in increased dipole moment. More succinctly said, AGoRaS is placing atoms beyond radii provided in the training data.

**Solution space**. An important aspect of VAEs and AGoRaS is the ability to map the latent probabilistic solution space of the problem using unsupervised learning and then sampling that solution space to generate new data. The latent space is hard to visualize conceptualized, especially as the size of the data being represented in it grows exponentially. As discussed in the previous section, it is the aim to demonstrate that the newly generated equations are filling in the solution space compared to the input and ultimately removing bias. Using a t-Distributed Stochastic Neighbor Embedding (t-SNE) plot, as seen in Fig. 4, it was possible to get a two-dimensional representation of what was happening in the high dimensional latent space. This is possible because t-sne plots use an unsupervised learning method of stochastic neighbor embedding to give high-dimensional data a single point on a two-dimensional grid. Figure 4 illustrates the t-SNE plot for AGoRaS. The blue circles represent the generated data set and the red circles represent the training data. The size of the sphere is proportional to the Gibbs free energy (Eq. (1)). The t-SNE algorithm groups data together based solely on their SMILES representations. After their placement on the t-SNE plot their thermodynamic properties were applied to the radius of each point. Figure 4A is a plot of the original 7000 equation in the

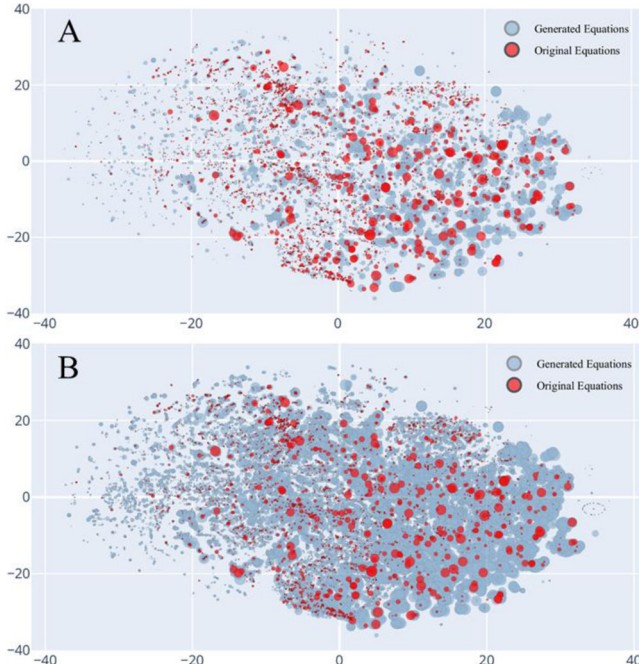

**Fig. 4 T-SNE plot of the training dataset (original equations) and the generated dataset (generated equations).** The size of the sphere is proportional to $\Delta G$ at standard conditions. **A** All 7000 equations from the training dataset and randomly sampled 7000 equations from the generated dataset. **B** 70,000 equations from the generated dataset.

training dataset with a sub-selection of an equivalent number of points for the generated dataset. Figure 4B is a similar plot with increased sampling of 70,000 generated equations. It is noted from these plots that generated data sets are not only filling between the training data but also extending out beyond the training data.

**Targeted reactions**. Another important application of AGoRaS is the ability to conduct targeted reaction searches. Researchers can utilize AGoRaS' ability to generate massive datasets comprised of new, unique reactions to search for a targeted molecular species and reactions[38]. To demonstrate this application, the original and generate datasets were searched for reactions containing $CO_2$ and reactions containing $CH_4$. The original dataset had approximately 150 unique reactions that contained $CO_2$, while the generated dataset had approximately 6000 reactions. For $CH_4$, the training dataset had approximately 700 reactions and the generated dataset had approximately 91,000 reactions. The generated data provided 40 times as many reactions to examine further from a thermodynamic perspective. As discussed above, AGoRaS is able to generate intermediate molecular species and new molecular species beyond the descriptions of the training dataset. Figure 5 is a histogram of the selection of both $CO_2$ and $CH_4$ for the training dataset and the generated dataset. The selection for both cases was down-selected to Gibbs reaction energies, ranging between $\Delta G \pm 5$ eV. Again, the network is able to avoid the bias of the training dataset demonstrating the utility of this approach compared to other approaches.

## Discussion
In this article, a new method for synthetic data generation of chemical equations known as AGoRaS was introduced. This article focused on AGoRaS' application for only gas-phase reactions but has applicability well beyond. The aim of AGoRaS was the avoidance of bias inherent to all training data in the

generation of continuous synthetic datasets. In this study, AGoRaS was trained on a core dataset comprised of only ~7000 reactions and 2000 molecular species and was able to generate 7,000,000 reactions with 20,000 molecular species. This is an extremely high return for such a small training set. The most exciting aspect of the results of AGoRaS was its ability to create a large quantity of new molecular species and overall reactions that were stable. AGoRaS implemented a unique approach of selecting from the latent space to generate the new reactions. This is a different approach as the typical usually involves retrosynthesis of existing reactions. An advantage of this approach is that the VAE gathers the knowledge of the physics and chemistry that it is trained on. This allows AGoRaS to generate new molecular species and reactions beyond the size and description of the training data.

It should not be overlooked that the generated results of AGoRaS have not been experimentally synthesized. Other studies in the past have noted that many molecules derived from generative networks cannot be synthesized[39]. This is even in light of the generated molecules scoring well on various quantitative benchmarks. While this may be the case for some of the molecules generated in the study. By using a semi-empirical technique, which has been calibrated with experiments, there is added

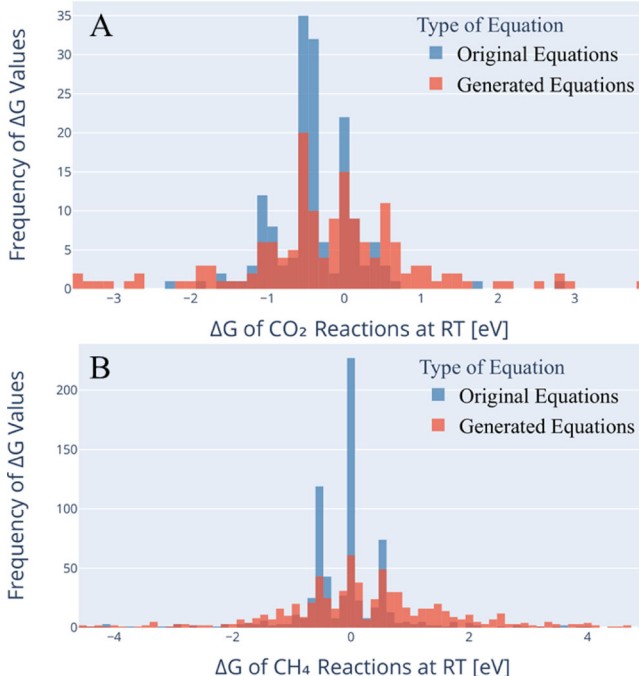

**Fig. 5 Histogram of the Gibbs energy at room temperature for specific molecular species. A** 168 unique equations containing $CO_2$ selected from both the training dataset (original equations) and generated dataset (generated equations). **B** 688 unique equations containing $CH_4$ selected from both the training dataset and generated dataset.

confidence many of these reactions and molecular species can be synthesized Moreover, this study attempted to mitigate this risk by checking all molecules against existing databases such as RDKit and Pubchem, in addition to conducting the semi-empirical calculations. Reactions that did not pass this check with both RDKit and the semi-empirical technique were eliminated from the final dataset.

To demonstrate the utility of AGoRaS the generated reaction equations were filter for reaction containing $CO_2$. Table 1 is a summary of a select number of these reactions that demonstrate the complexity of the reactions that were discovered. Note the equations displayed in Table 1 were not part of the training data.

One of the limitations of AGoRaS is that the synthetic generated data still may share similar distributions and biases with the training data. However, this approach is an improvement on the current techniques as it has the ability to expand beyond the molecular description of the training dataset. As the AGoRaS is trained on larger datasets and potentially fed back generated data, the bias will diminish. It would be trivial to analyze the training dataset for biases and introduce synthetic equations that balance out the biases. The network could be retrained and sampled again to check for biases. This process could be repeated until the network is no longer producing biased outputs. While the uniqueness of the dataset can be satisfied, the completeness of the dataset will never be achieved due to the continuous nature of the data. However, this provides a tool for achieving completeness over a bound design space and satisfying the local continuity. But this may be acceptable for engineering applications where absolute knowledge is not necessary.

## Methods
The AGoRaS method was developed to synthetically generate chemical equations without the need for human interaction. As discussed above, a pipeline or workflow was created to accomplish the following steps outlined in Fig. 2. The motivation behind having a pipeline set up was so that new data could easily be ingested into the pipeline to quickly and effectively train AGoRaS on a variety of data. The use of data pipelines and their effect on code quality and robustness is a well-studied topic[40–42]. This ability takes a considerable amount of data preconditioning out of the hands of the user, which will allow AGoRaS to be more accessible to a non-data scientist. It will instead allow chemists and engineers to utilize the network using their own data. This has historically been a problem, where the lack of data science skills and the steep learning curve of network development can hold back a field from utilizing the power of artificial intelligence[43].

**Data collection**. The chemical equations used to train this network were provided by the NIST chemical kinetics database[36]. The data was received from them as two separate CSV files. One file contained a column of 15,000 id numbers and several other columns that all had different naming conventions. These included chemical formula, hill sorted formula, and IUPAC name. The second CSV file contained rows corresponding to different chemical reactions in the dataset. Each of the reactions was represented by the id numbers from the first CSV. The first CSV was read into python using a pandas dataframe[44] where each possible naming convention was a different column.

**Convert data to SMILES notation**. SMILES was chosen to be used as the ground truth representation of chemical species due to its ability to represent species with the same chemical formula but different structures uniquely[45]. SMILES is widely used in chemical artificial intelligence and has shown excellent results from other

**Table 1 Selected results from the AGoRaS network that illustrates the SMILES notation and equivalent chemical equation representation.**

| SMILES equation | Chemical equation | $\Delta G$ (eV) |
|---|---|---|
| [O]=[C]=[O] + 2[H][H] → [H][C]([H])=[O] + [H][O][H] | $CO_2 + 2H_2 \rightarrow CH_2O + H_2O$ | 0.204 |
| 1[H][C] + 2[H][O][H] → 4[H][H] + 1[O]= [C]=[O] | $CH_4 + 2H_2O \rightarrow 4H_2 + CO_2$ | 0.047 |
| [H][O][C]([H])([H])[C](=[O])[O][C]([H])([H])[H] → [H][O][C]([H])([H])[C]([H])([H])[H] + [O]=[C]=[O] | $C_3H_6O_3 \rightarrow C_2H_6O + CO_2$ | −0.486 |

The rightmost column is the associated predicted Gibbs energy from the semi-empirical calculation.

techniques[46–50]. Despite its popularity, there is no easy and open-source way to convert different chemical names into SMILES. The pipeline utilizes several different techniques to convert the different names into SMILES notation.

Since the available names of the 15,000 species were uploaded into a pandas DataFrame, it was possible to check each one on an open-source website for conversion. In the end, two main servers were used for conversions due to their large repositories and their trustworthy data sources: the CADD Group Chemoinformatics Tools and User Services[51] and PubChem[34]. Each possible name of the 15,000 different species was put through both PubChem and the CADD groups databases.

**Preprocessing the data**. Once all species were converted, a comparison was done between both databases results and any species that could not be converted, or that was not in agreement between the two was eliminated from the pool of valid species. This was part of our attempts to be both rigorous and autonomous in our species data pipeline. It would have been possible to manually go through the different disagreements and correct them, but this fell outside of the aims of this study. Since one of the main goals was reproducible, this was deemed the best strategy. A tertiary check of the species was performed; each of the species was checked for stability and feasibility using the RDkit package in Python[33]. RDkit is another widely used and well-verified package for computational chemistry that is often paired with SMILES notation machine learning algorithms[1,3,9,46,49,50,52].

Once the number of available species had been reduced, the remaining species could be mapped to the CSV file containing the actual reaction equations. Any equation that did not have a SMILES identifier for a given species was eliminated from the dataset. An additional check was performed where each of the reaming equations was checked to be sure they were balanced. This was done to catch any equations that may have had incorrect ids for reactants or products. A further criterion was placed on the equations, where all equations with more than three species on the product or reactant sides of the equations were eliminated. This was done to facilitate the convergence of the VAE during training by having similar length character vectors and the limited availability of training data with more than three species. Research by Dwarampudi et al. and Prusa et al. demonstrated that padding the LSTM and neural networks can cause instability and result in poor network performance[53]. To this end, it was decided that using character-level embedding offered an advantage over word-level embedding[53,54]. This would allow for the generative network to not just use the same species it had been trained with, but instead allow for it to generate new species based on information gained in the training process. The embedding was done using TensorFlow's built-in embedding techniques and was based on a universal alphabet created from all the equations[55]. Additionally, Gaspar et al. demonstrated how molecule embedding can mirror that of NLP embedding. They go on to demonstrate how improvements can be made to the predictive power of machine learning models by formulating the inputs into sequence embeddings[56]. This is an extremely interesting avenue to pursue and could lead to an increased number of attractive and viable reactions for AGoRaS.

**AGoRaS VAE structure**. AGoRaS takes in a vector representation of length $n$, where $n$ is the maximum length character-level representation of any equation in the training dataset. This vector is then fed forward into a TensorFlow embedding layer that projects the input into a higher dimensionality. This is an important step because the projection fits numeric values to a high dimensionality space, removing the intrinsic value from the values themselves. This is done because in the two-dimensional space, the numeric values have no intrinsic value (i.e. five is not greater than six they just represent two different characters). The projected vectors are then fed into a bidirectional LSTM (BiLSTM) layer with a recurrent dropout of 0.2. The mean and log variance is created from the output of the BiLSTM layer. A sampling function is used to randomly sample this solution space based on the mean and log variance. The network then decodes the sampled solution space using a RepeatVector layer that is wrapped around the output of the latent space, thus turning it into a tensor vector that an LSTM layer could read. The Repeat-Vector is fed as an input into an LSTM layer, whose output is projected into a vector of length $n$. The output of this projection is what is used to calculate the loss of the network. AGoRaS uses a sequence-to-sequence style loss function common to variation autoencoders. The metric used for monitoring the network during training is also the standard kl loss. The network was trained for 500 epochs using a batch size of 25, an embedding dimensionality of 500, and a latent dimensionality of 350. The $kl$ weight used was 0.1 and the activation function was a softmax function. The optimizer function was Adam and the learning rate was $1 \times 10^{-5}$.

**Training AGoRaS**. Once the training data had been cleaned and embedded in a numerical format readable by a neural network architecture, it is split into three sections. The training set utilized 70% of available data. The validation set was 20% of available data, and test data was 10% of the data. Even though AGoRaS utilizes a generative technique, it can still be validated using traditional methods. The VAEs was validated by testing the ability of the network to encode the validation set and decode it back to the original string construction with no loss of information.

**Autonomously sampling the latent space**. Once a trained network has been created, it is possible to construct a sampler that directly interfaces with the latent representation. This is shown in Fig. 1B where the neural net layers responsible for the encoding are removed from the network. It is possible to then start sampling the different latent representations randomly using a randomized point for each element and having the decoder part of the network constructing equations. The decoder takes the randomized points and applies the learned weights to construct new equations. Due to the probabilistic nature of the latent representations, it is possible to take an almost unlimited number of sample points and continue to generate new equations. Of course, there would be diminishing returns on this, as there are only so many chemically feasible equations possible.

**Validating generated equations**. The methodology for determining the chemical validity of equations was extremely like that of the data cleaning process. The first step was to eliminate duplicate equations. The second step was to check if the generated equations were balanced. This offered a computationally inexpensive way to cut down on the number of equations present in the generated dataset. The third step was to check each species for chemical validity using RDKit, where any physically or chemically unstable species should be rejected by the software[33]. This is a common practice when using a neural network with generated SMILES species[9,46,47,50,57].

**Perform semi-empirical calculation**. Using the SMILES notion provided in the generated dataset output, a custom Pipeline Pilot protocol[58] was written that would take the SMILES entry and convert it to an atomistic description. Once the data was converted to an atomistic description, a semi-empirical calculation was conducted to predict the thermodynamic properties. Pipeline Pilot[58] developed by Dassault Systems is a powerful tool capable of manipulating and analyzing large quantities of scientific data by using automated processes. The input to Pipeline Pilot was a CSV file containing the SMILES description for all molecular species. The output was the thermodynamic data in a CSV format. Over 7,000,000 chemical equations were processed through the automated Pipeline Pilot protocol to generate the thermodynamics data for this study.

The semi-empirical calculation is a well-developed method that supersedes many of the more rigorous quantum chemistry methods employed in similar studies. Semi-empirical calculations are a good starting point for many of these more rigorous density functional theory techniques. The approximations for the complex interactions in the Hamiltonian are accounted for using empirical parameters tuned to reproduce experimental results. The semi-empirical method determines the molecular wavefunction, which in turn provides a prediction of the thermodynamic properties. The molecular wavefunction is constructed using a linear combination of atomic orbital (LCAO) method. The molecular orbitals are determined based on a linear combination of Slater-type atomic orbitals. The semi-empirical method uses Slater functions by evaluating the two-electron integral via a multipole approximation[35].

The semi-empirical model that was implemented in the Pipeline Pilot protocol was based on the Materials Studio provided VAMP software package[35,37]. Geometry optimization was conducted with a range of Hamiltonian models that include a diatomic differential overlap (NDDO) and PM6 Hamiltonian, Auto multiplicity, and a spin state unrestricted Hartree-Fock (UHF), restricted Hartree-Fock (RHF), or annihilated unrestricted Hartree-Fock (A-UHF). Several spin states were tested based on convergence. A Paulay/IIS convergence scheme with a convergence energy tolerance of $2 \times 10^{-4}$. The approach began with the most restrictive spin state Hamiltonian model and if convergence was not met, a less restrictive Hamiltonian was tried. Once convergence was achieved for one of the models, the thermodynamics information and total dipole moment were outputted from the Pipeline Pilot script to a CSV file.

The Pipeline Pilot protocol script conducted a series of preparation steps before the semi-empirical calculation. After data was read in using the SMILES format the SMILES formula was checked for consistency, followed by constructing the atomistic description of the molecule to provide an initial geometry. The molecule was generated from the SMILES, centered in the domain, hydrogen atoms were added to achieve the correct overall charge, and the molecule was cleaned to provide a initial atomistic geometry. The cleaning step conducted a quick empirical electrostatic relaxation of the structure to refine the initial geometry. The structure was provided to a programmed series of VAMP calculations starting with the most rigorous spin state, as stated above.

**Calculation of thermodynamics properties**. Once the quantities of interest for each molecular species had been calculated using the semi-empirical model, the thermodynamics for each reaction were determined. For each molecular species, the entropy, enthalpy, and dipole moments were provided at the standard state at room temperature. The Gibbs free energy for each reaction was calculated using Eq. (1). Any overall reaction Gibbs energy that exceeded ±5 eV was eliminated due to concern with chemical stability. Once this step was completed, the equations were deemed safe to publish and for engineers to interpret.

## Data availability
Machine learning model is available on github repository with accompanying filtered experimental datasets and semi-empirical datasets. Unfiltered datasets and Pipeline Pilot protocols used to generate filtered datasets can be requested from corresponding author.

## Code availability

AGoRaS Git: https://github.com/Dr-Musho-Research-Group/AGoRaS-VAE.git.

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

## Acknowledgements

T.M. would like to recognize the partial funding support from the National Science Foundation (NSF) Award #1709568. R.T. would also like to recognize the partial funding of the Department of Energy (DOE) ORISE Fellowship Program. T.M. and R.T. would like to recognize Dushyant Shekhawat at NETL for the chemical engineering application discussion. R.T. would also like to recognize the utilization of the HPC system Joule at NETL. R.T. and T.M would like to recognize the utilization of the HPC system Thorny Flat at WVU, which was constructed by NSF Major Research Instrumentation (MRI) Grant Award #1726534.

## Author contributions

R.T. contributed to ML model development and writing of manuscript. T.M. contributed to ML model development, semi-empirical data synthesis, and writing of manuscript.

## Competing interests

The authors declare no competing interests.
