## [Peer Review File · Communications Chemistry]

Reviewers' comments:

Reviewer #1 (Remarks to the Author):

The authors have developed a VAE method for data augmentation of chemical reactions.

The study is well written and executed.

However, there are issues with the openness and general interest of the study.

As the author states this is not a comparison study between different ML methods, instead it is a method to augment a reaction data set. For that to be interesting for the scientific community the source code needs to be released with the submission and not eventually at a later stage.

Both the training set and the large set of generated reactions needs to also to be made publicly available to make this a meaningful study of scientific interest.

Finally, I don't think the study is of enough general scientific interest to warrant a publication in Communications Chemistry. In the Nature family I would recommend Nature Scientific Data or from other publishers a cheminformatics journal like Journal of Cheminformatics

Reviewer #2 (Remarks to the Author):

The authors used a deep learning model (VAE) to generate new chemical reactions to synthetically generate continuous datasets. This is something meaningful and valuable. However, some issues need to be addressed before publication.

(1) As far as I know, VAE is an unsupervised learning model, rather than semi-supervised model.

(2) It is not appropriate to name the method ChemNet, there is a commercial site called ChemNet: <http://www.chemnet.com/>.

(3) A major problem is that the authors should clarify the novelty of this study. Some published works have using AI methods to improve reaction prediction and optimize chemical synthesis. Please refer to: Nat. Chem. 12, 509–510 (2020); Nature 590, 89–96 (2021).

(4) The authors were trying to solve the problem of data discontinuity by using the method of data generation. There are some strategies that can address the challenge of costly and scarce experimental data in chemical engineering, such as latent space training and other unsupervised learning methods. The authors can add discussions where appropriate. Please refer to: J. Phys. Chem. A 2019, 123, 19, 4295–4302; Cell Rep Phys. Sci. 1, 100269.

(5) I do think the subtitles of Section II are confusable, especially for Part B and C. More, the authors can make a workflow for the whole study to make it easier for the readers to understand.

(6) Part C in Section II was not clearly stated. According to my understanding, the authors generated 7,000,000 valid equations using VAE based on the known 7,000 equations, and then selected 7,000 new equations to compare trained and generated datasets, it was reasonable. However, 3,000,000 equations were also mentioned, how did they come from and what do they do?

(7) Some parameters should be given clearly. Such as the layers of VAE, the perplexity of t-SNE plot. Please refer to: J. Am. Chem. Soc. 2018, 140, 32, 10158–10168

(8) I suggest that the authors highlight the new generated reactions in order to generalize them to practical applications, 3–5 representative cases are enough.

(9) Generative Adversarial Networks (GAN) is also a popular generation model, which may help to address some limitations of VAE in this study; the authors can supplement more discussions on GAN in Introduction or Discussion.

Reviewer #3 (Remarks to the Author):

Review for the article titled "Autonomous Molecular Synthesis using Deep Learning"
by Robert Tempke and Terence Musho

In this paper the authors rightly emphasise the importance of data bias in machine learning and optimisation models applied to chemistry. The authors then propose an automatic approach to select a set of balanced chemical reactions compatible with SMILES writing. From this set of 7000 reactions, a variational autoencoder is trained and then used to generate new balanced chemical equations.

After several readings, my opinion is very divided and at least major points need to be addressed. On the one hand I find the bibliography partial. This does not allow me to situate this study in the state of the art (see below). On the other hand, the use of the term "continuous data set" in the abstract is problematic in my opinion. The data set is a set of chemical reactions associated with a free energy. Do the authors consider two equations with the same free energy as neighbours? In the article, the reader is left with the impression that the continuity is in latent space. But the article does not discuss latent space enough, I think.

1. Figure 4 shows that the latent space does not appear to be particularly sorted by free energy. The authors assume that the autoencoder learns chemistry and physics but do not demonstrate this further. There is no discussion of the quality of the new equations generated. What form of neighbourhood does the autoencoder propose? Do the authors not show enough interest in the new reactions generated? I think this is a crucial aspect. Does any balanced equation with a calculated free energy count as a chemical reaction? What about the different outcomes?

2. Concerning the bibliography. The introduction starts with the interplay of machine learning and DFT. And for this very broad research 2 specific articles are cited when some recent reviews could have been more representative.

Concerning the predictions of chemical reactions or retrosynthetic routes, I am surprised that the work of Coley, Segler or Engkvist is not cited at all.

3. Concerning the calculations. The article indicates a DFT approach in Fig 2 and then in the text mentions a semi-empirical DFT approach. But in section H, it seems that it is a semi-empirical HF method that has been used.

Some small points:

- " This ensured that each chemical reaction in the dataset was unique and could be balanced." Not in the case of the example of Fig 1B or it should mention that the H are implicit all along and that could have a huge impact on the quality of the newly generated reactions if H atoms can appear and disappear no ?

- Check for the missing references like RDKit at some point of the text.

- "non-intermediate species" ? What do the authors mean ? transition states ?

- "A further criterion was placed on the equations, where all equations with more than three species on the product or reactant sides of the equations were eliminated. This was done to facilitate the convergence of the VAE during training by having similar length character vectors." SMILES characters lengths depends on the size of the molecules more than on the number of species no ?

Possible typos:

"This especially true in the field of chemistry"

"This study aims to use the utilize the predictive abilities of deep learning to synthetically generate a chemical reaction dataset "

"This is a unique approach in relies on artificial intelligence to generate"

"Inorganic chemistry, Kayala, and Baldi show how machine learning can be utilized to"

"A unique approach applied in this research in the application to chemistry is that instead of each latent variable being encoded with a discrete value, as with a traditional autoencoder, instead, the latent variable has an associated probability distribution."

"pupchem [25]"

"The dipole moment was also select because"

"The second CSV file contained [x] rows"

Cordially

Review One's Comments:

"Both the training set and the large set of generated reactions needs to also to be made publicly available to make this a meaningful study of scientific interest."

We have made the source and datasets to reproduced results available on Github. We also added a sentence to the data availability section.

"Finally, I don't think the study is of enough general scientific interest to warrant a publication in Communications Chemistry. In the Nature family I would recommend Nature Scientific Data or from other publishers a cheminformatics journal like Journal of Cheminformatics"

We tried to make the discussion a little more general including the abstract.

Review Two's Comments:

"(1) As far as I know, VAE is an unsupervised learning model, rather than semi-supervised model."

We removed the saying semi-supervised from the manuscript.

"(2) It is not appropriate to name the method ChemNet, there is a commercial site called ChemNet: <http://www.chemnet.com/>."

We renamed the network, AGoRaS which stands for, autonomous generation of reactions and species-VAE. Nice catch.

"(3) A major problem is that the authors should clarify the novelty of this study. Some published works have using AI methods to improve reaction prediction and optimize chemical synthesis. Please refer to: Nat. Chem. 12, 509–510 (2020); Nature 590, 89–96 (2021)."

The novelty of the approach is really sampling the latent space. You're correct we are not the first to answer this problem but our approach has some clear advantage. We add several paragraphs to the manuscript based on this comment to support this claim. We have also supported the unique ability generate larger molecules that were not in the training dataset.

"(4) The authors were trying to solve the problem of data discontinuity by using the method of data generation. There are some strategies that can address the challenge of costly and scarce experimental data in chemical engineering, such as latent space training and other unsupervised learning methods. The authors can add discussions where appropriate. Please refer to: J. Phys. Chem. A 2019, 123, 19, 4295–4302; Cell Rep Phys. Sci. 1, 100269."

We have added additional three additional paragraphs to the manuscript following your provided references. You're correct we are not the first to try to solve this problem.

"(5) I do think the subtitles of Section II are confusable, especially for Part B and C. More, the authors can make a workflow for the whole study to make it easier for the readers to understand."

Changed Section II part B to Pipeline for AGoRaS-VAE Training and Validation, changed part C to Validating Generated Equations using Physical Quantities. The workflow for the study is shown in Figure 2.

"(6) Part C in Section II was not clearly stated. According to my understanding, the authors generated 7,000,000 valid equations using VAE based on the known 7,000 equations, and then selected 7,000 new equations to compare trained and generated datasets, it was reasonable. However, 3,000,000 equations were also mentioned, how did they come from and what do they do?"

Replaced the 3,000,000 with 7,000,000.

"(8) I suggest that the authors highlight the new generated reactions in order to generalize them to practical applications, 3–5 representative cases are enough."

Added a table (Table 1) with equations in SMILES notation, chemical equations and gibbs free energy at room temperature.

(9) Generative Adversarial Networks (GAN) is also a popular generation model, which may helpful to address some limitations of VAE in this study; the authors can supplement more discussions on GAN in Introduction or Discussion.

We added some additional GANs discussion to the introduction and added additional references.

Reviewer Three's Comments:

"On the other hand, the use of the term "continuous data set" in the abstract is problematic in my opinion. The data set is a set of chemical reactions associated with a free energy. Do the authors consider two equations with the same free energy as neighbours? In the article, the reader is left with the impression that the continuity is in latent space. But the article does not discuss latent space enough, I think."

We agree with this. We added some additional discussion in the narrative to support our claims of a continuous dataset.

"1. Figure 4 shows that the latent space does not appear to be particularly sorted by free energy. The authors assume that the autoencoder learns chemistry and physics but do not demonstrate this further. There is no discussion of the quality of the new equations generated. What form of neighbourhood does the autoencoder propose? Do the authors not show enough interest in the new reactions generated? I think this is a crucial aspect. Does any balanced equation with a calculated free energy count as a chemical reaction? What about the different outcomes?"

The only information given to AGoRas-VAE is the encoded SMILES string and the only output is an encoded SMILES string, in which hydrogen atoms are implicit

The t-sne algorithm grouped data together based solely on their SMILES representations. It was only after that their thermodynamic properties were applied to the radius of each point. This helped to show us that the real and generated equations had almost identical distributions of thermodynamic properties in the compressed space of the t-sne. This means that the network is learning how to generate equations with balanced thermodynamic quantities without explicitly needed to be trained on it.

Added Table 1 to show some actual generated equations.

"2. Concerning the bibliography. The introduction starts with the interplay of machine learning and DFT. And for this very broad research 2 specific articles are cited when some recent reviews could have been more representative. Concerning the predictions of chemical reactions or retrosynthetic routes, I am surprised that the work of Coley, Seqler or Engkvist is not cited at all."

Thanks for the comment. We have added some additional reference point out by this comment and other comments. Specifically, we have added a paragraph discussing Coley.

"3. Concerning the calculations. The article indicates a DFT approach in Fig 2 and then in the text mention a semi-empirical DFT approach. But in section H, it seems that it is a semi-empirical HF method that as been used."

You're correct. There were some conflicting descriptors used here. This has been corrected in the manuscript. The method was a semi-empirical technique based on HF method.

“Some small points:

- " This ensured that each chemical reaction in the dataset was unique and could be balanced." Not in the case of the example of Fig 1B or it should mention that the H are implicit all along and that could have a huge impact on the quality of the newly generated reactions if H atoms can appear and disappear no ?”

The only information given to AGoRas is the encoded SMILES string and the only output is an encoded SMILES string, in which hydrogen atoms are implicit. We check to make sure that the equations are balanced, and the species are real which eliminates that risk.

“Check for the missing references like RDKit at some point of the text.”

Added reference for RDKit.

"non-intermediate species" ? What do the authors mean ? transition states ?

We replaced this terminology in the manuscript. What we meant by non-intermediate is the opposite of the subspecies or the retrosynthesized species.

"A further criterion was placed on the equations, where all equations with more than three species on the product or reactant sides of the equations were eliminated. This was done to facilitate the convergence of the VAE during training by having similar length character vectors." SMILES characters lengths depends on the size of the molecules more than on the number of species no ?

Yes, you are correct, SMILES depends on the size of the molecules. But since we only have maybe 5-10 equations with 4 or 5 species we didn't have enough data to train the network on equations of that length. We added a sentence to this effect in the manuscript.

REVIEWERS' COMMENTS:

Reviewer #2 (Remarks to the Author):

The authors have addressed all the comments and made the changes accordingly to meet the journal requirements.

Reviewer #3 (Remarks to the Author):

Review for the revised version of Autonomous Molecular Synthesis using Deep Learning by Robert Tempke and Terence Musho.

The introduction has been extensively revised. The context and scope of the study are better defined. The methodology is also better explained.

I still noticed a few minor typos which I list below. But apart from these minor remarks, I give my approval for the publication of this article.

Kindly,

suspected typos:

- "In, this study, that the correct prediction of the reactions was reached but only because of a reaction bias."
- "This means that these deep learning techniques can interpreted between existing data points"
- "There work uses both real and density functional theory predicted models to predict properties"
- "Take the work form Zhang et al"
- "an encoded SMILES sring"
- "datasets, both the original and generate, have a single peak near zero"
- "the original and generate datasets were searched for reactions containing CO2"
- "It is noted from Figure 5 that both the new equations generated that contained both select molecular species."
- "The output was the the thermodynamic data"
- "From a molecular point of view, this has the opposite trend of entropy, as being the larger the molecule the more likely the charge will be neutral for the molecule and have a negligible dipole moment." No typo here, however the VAE could generate easily push pull compounds. So this assumption may be exaggerated.
- "names of the 1,500 species were uploaded into a Pandas data frame" Isn't it 15k?

- "Over 7,000,000 molecular species were processed through the automated protocol to generate the thermodynamics data for this study" equations ?

Reviewer #3 Comments:

There are corrections that were made in the manuscript.

1. Reviewer Comment:

There work uses both real and density functional theory predicted models to predict properties

a. Author Response:

Their research uses both experimental and density functional theory predicted models to predict properties.

2. Reviewer Comment:

Take the work form Zhang et al

b. Author Response:

Modified sentence: Take the research from Zhang et al

3. Reviewer Comment:

datasets, both the original and generate, have a single peak near zero

c. Author Response:

Modified sentence: Therefore, as seen in Figure 3, both the original and generated datasets have a single peak near zero

4. Reviewer Comment:

names of the 1,500 species were uploaded into a Pandas data frame" Isn't it 15k

d. Author Response:

Modified sentence: Names of the 15,000 species were uploaded into a Pandas Dataframe.

5. Reviewer Comment:

"Over 7,000,000 molecular species were processed through the automated protocol to generate the thermodynamics data for this study" equations ?

e. Author Response:

Modified sentence: Over 7,000,000 chemical equations were processed through the automated protocol to generate the thermodynamics data for this study

These were some additional corrections that were in the clean version but not the track changes version in the revision. These have all been corrected.

1. Reviewer Comment:

In, this study, that the correct prediction of the reactions was reached but only because of a reaction bias.

a. Author Response:

Modified sentence: It was claimed that the correct prediction of the reactions was achieved but only because of a resulting reaction bias

2. Reviewer Comment:

This means that these deep learning techniques can interpreted between existing data points

a. Author Response:

Modified sentence: This means that these deep learning techniques can interpret between existing data points to generate new synthetic derived points that share the same characteristics as the original dataset

3. Reviewer Comment:

an encoded SMILES string

a. Author Response:

Modified sentence: an encoded SMILES string

4. Reviewer Comment

The output was the the thermodynamic data

a. Author Response:

Modified sentence: The output was the thermodynamic data

This one I can't find in the paper?

1. Reviewers comment:

It is noted from Figure 5 that both the new equations generated that contained both select molecular species.

a. Author Response:

This sentence was removed.

2. Reviewers Comment:

the original and generate datasets were searched for reactions containing CO2

a. Author Response:

This sentence was corrected.

1. Reviewers Comment:

-"From a molecular point of view, this has the opposite trend of entropy, as being the larger the molecule the more likely the charge will be neutral for the molecule and have a negligible dipole moment." No typo here, however the VAE could generate easily push pull compounds. So this assumption may be exaggerated

a. Author Response:

This sentence has been reworded and clarified in the paper.